# Using EmPalPed—An Educational Toolkit on Essential Messages in Palliative Care and Pain Management in Children—As a Strategy to Promote Pediatric Palliative Care

**DOI:** 10.3390/children9060838

**Published:** 2022-06-06

**Authors:** Ximena García-Quintero, Angélica Claros-Hulbert, María Elena Tello-Cajiao, Jhon Edwar Bolaños-Lopez, María Isabel Cuervo-Suárez, Martha Gabriela García Durán, Wendy Gómez-García, Michael McNeil, Justin N. Baker

**Affiliations:** 1Fundación Valle del Lili, Centro de Investigaciones Clínicas, Cali 760032, Colombia; maria.cuervo@fvl.org.co; 2Division of Quality of Life and Palliative Care, St. Jude Children’s Research Hospital, 262 Danny Thomas Place, MS #260, Memphis, TN 38105, USA; justin.baker@stjude.org (J.N.B.); michael.mcneil@stjude.org (M.M.); 3Faculty of Health Sciences, Universidad de la Sabana, Chia 250001, Colombia; angelicadelm33@hotmail.com; 4Palliative Care Department, Instituto Nacional de Cancerología, Bogotá 111511, Colombia; 5Internal Medicine, Department, Universidad Libre, Cali 760043, Colombia; maria-telloc@unilibre.edu.co; 6Dirección de Investigación y Desarrollo, Centro de Biociencias, Seguros SURA Colombia, Medellín 050021, Colombia; jebolanos@sura.com.co; 7Faculty of Health Sciences Department, Clinical Medical Science, Universidad Icesi, Cali 760031, Colombia; 8Psycho-Oncology Service, Hospital Pediátrico de Sinaloa “Dr. Rigoberto Aguilar Pico”, Culiacán 80200, Sinaloa, Mexico; psicooncologiahps@gmail.com; 9Dr. Robert Reid Cabral Children’s Hospital, Santo Domingo 10107, Dominican Republic; emogenes@yahoo.com

**Keywords:** Palliative care, education, healthcare professional, pain assessment, low-income countries, middle-income countries, physicians

## Abstract

Background: Most children needing palliative care (PC) live in low- and middle-income countries. In Colombia, pediatric palliative care (PPC) knowledge among healthcare professionals (HCPs) is lacking as PPC is not included in the educational curricula of healthcare programs. Therefore, specific training that improves knowledge of HCPs and access to PC for children and their families is needed. To address this gap, we organized and conducted the Essential Messages in Palliative Care and Pain Management in Children (EmPalPed), an educational toolkit to increase awareness and promote essential knowledge in PPC for low- and middle-income countries. Methodology: The EmPalPed toolkit consisted of a 5-h virtual workshop with small working groups of HCPs caring for children with life-threatening conditions such as cancer. The toolkit was organized along five key domains: (1) PC as it relates to the concept of quality of life (QoL), (2) effective communication, (3) addressing pain management as a top priority, (4) providing end-of-life care, and (5) access to high-quality PC as a fundamental human right. The workshop activities included different educational strategies and tools (e.g., a pocket guide for pain assessment and management, a PPC booklet, a quick guide for communicating bad news, role playing, and discussions of clinical cases). Results: A total of 145 HCPs from 22 centers were trained. The post-test analysis for HCPs showed that attitude and knowledge about communication (*p* < 0.001), pain assessment (*p* < 0.001), first-line opioid of choice in children (*p* < 0.001), and palliative sedation (*p* < 0.001) had positive and statistically significant changes from the pre-test analysis. Discussion: This study supported the notion that the EmPalPed educational toolkit is an effective mechanism for raising awareness regarding PPC as well as providing training in many of the key aspects of PPC. The EmPalPed training approach should be studied beyond this setting, and the impact should be measured longitudinally.

## 1. Introduction

Palliative care (PC) is a field of medicine that provides care for patients experiencing serious illness and their families at all stages of their disease trajectory. The World Health Organization (WHO) states that PC is a human right [1]. Integration of palliative care into the ongoing care of both adults and children can improve quality of life and positively impact other patient-reported outcomes as well [2].

Despite the benefits of pediatric palliative care, there is an extensive list of barriers to the early integration of PC in the disease trajectory [3]. The barriers include prognostic uncertainty, time constraints, discomfort of health care staff in providing pediatric palliative care (PPC), and a lack of knowledge, experience, and team support [4,5,6,7,8,9,10]. Part of the suffering of children with palliative needs is caused by pain, some of the barriers are the inadequate basic education in all relevant health disciplines such as doctors, nurses, therapists, psychologists, to perform a multimodal approach and management of pain [11]. Most children (98%) in need of PPC live in low- and middle-income countries [2], and Latin America has one of the highest percentages of children in need of oncologic PPC [12,13]. 

According to the Latin American Atlas of PC, only 3 of the 57 medical schools in Colombia include PC in their curricula on a mandatory basis [14], and among 359 pediatricians and residents nationwide, only 13% had knowledge on the subject [13]. This lack of knowledge about PC likely perpetuates the suffering of children and their families [15,16]. Therefore, education on PC for HCPs caring for children with life-threatening and life-limiting conditions is necessary to increase the provision of this service to this population and their families. 

For these reasons, we developed a toolkit and workshop to promote advocacy for PPC and educate Spanish-speaking professionals about PPC. The aim of this paper is to describe the development of the toolkit and analyze its short-term effect on professionals. The educational toolkit material was based on the World Health Assembly resolution WHA 67.19, which called on WHO and member states to improve access to palliative care as a core component of health systems. One of the principles of WHA 67.19 is the strengthening of palliative care education for health professionals through the development of integrated palliative care guidelines and tools across disease groups and levels of care. 

## 2. Methods 

### 2.1. Workshop and Toolkit Development

The EmPalPed educational toolkit material was developed through an iterative process of consensus-building by a multidisciplinary palliative care expert group consisting of a pediatrician, a family physician, a social worker, and a psychologist with advanced training in pediatric palliative care, from the University Hospital–Fundación Valle de Lili. This institution is a center of excellence in palliative care in Colombia that provides consultation, education, and support to primary and secondary level practitioners (Table A1). The educational toolkit was thought and designed for all health workers involved in clinical practice, interested in learning PPC despite occupational categories (General Practitioners and specialty physicians, psychologists, social workers, and other healthcare professionals such as nursing assistants, nurses, physical therapists, speech therapists, pharmaceutical chemists). The material consisted of a set of resources: 1. Booklet with an overview of PPC, 2. A pocket reference for pediatric pain assessment and management, and 3. Educational cards about communication strategies to deliver bad news; and a synchronous virtual workshop titled “Essential Messages in Pediatric Palliative Care and Pain Management”.

The educational workshop was facilitated by a multidisciplinary and multicultural team that was also part of the educational toolkit development. Each of the facilitators developed different topics according to their knowledge and experience. The pedagogical strategy used to develop the workshop was constructivist, which implies that the activities developed allow the students to access their experiences and beliefs, which modifies their previous knowledge in light of the applied contents of the course [17].

The workshop addressed five key topic areas for the region in a 5-h activity: (1) palliative care as it relates to the concept of quality of life (QoL), (2) effective communication, (3) addressing pain management as a top priority, (4) providing end-of-life care, and (5) access to high-quality PC as a fundamental human right. Workshop activities included strategies such as a theoretical presentation of the topic, short videos, clinical case discussion, and role-playing (where participants were randomly assigned to act as a patient, family or healthcare professional and encouraged to pretend to be in a specific situation to determine physical, emotional, social, and family needs of the patient and family). In 2019, this education intervention was conducted face-to-face at the Fundación Valle del Lili University Hospital in Colombia. The SARS-CoV-2 pandemic necessitated that the subsequent workshops were conducted virtually and synchronously in 2020 across all Latin America (Table A2). The virtual and face-to-face workshop was conducted with the same content.

A brief screening demographic survey was conducted. Then, participants were surveyed about their attitude and knowledge via a “pre-test” (Table A3). Upon completing the educational intervention, the same questions were sent as a “post-test”. The test questions were intended to assess essential knowledge in palliative care, definition, patients who benefit from PC, communication strategies, pain assessment, pain management, and strengthen the concept of palliative sedation as a therapeutic strategy for patients.

### 2.2. Statistical Analysis

Nominal and ordinal variables were summarized as proportions. The group was divided into physician and non-physician individuals, and differences in the pre-test and post-test scores between the two groups were evaluated by the chi-squared test. When one of the expected values was ≤5, Fisher’s exact test was used. To analyze the effect of the workshop, we used McNemar’s test; values of *p* < 0.05 were considered to be statistically significant.

## 3. Results

A total of 145 HCPs from 22 hospitals were trained in 12 workshops held from June to December 2020. Approximately 97% of the participants were Colombian, and about 80% of them came from the city of Cali (Table A1). In addition, 70% were physicians (specialists in fields such as pediatric cardiology, pediatric intensive care, anesthesiology and pain, pediatric pulmonology, pediatric haemato-oncology, neonatology, pediatric neurology, and pediatric gastroenterology, or general practitioners), 29.6% were professionals from other branches of medical science (e.g., nursing, psychology, social work), and most were women (71.6%) (Table 1).

The group was divided into physician and non-physician HCPs, and we analyzed pre-test and post-test scores between the two groups. We found statistically significant differences between both groups’ performances in the pre-test questions assessing the appropriate scale for pain assessment (*p* < 0.002) and the opioid of choice (*p* < 0.001) in pediatric patients (Table 2).

The post-test analysis (Table 3) for physicians showed that the areas of attitude about communication (32% change, *p* < 0.001), pain assessment (81.5% change, *p* < 0.001), first-line opioid of choice in children (16.7% change, *p* < 0.001), and palliative sedation (72.7% change, *p* < 0.001) had positive and statistically significant percent changes from the pre-test analysis. Similar to the physician group, in other healthcare professionals, the post-test study showed that the areas of communication attitude (percentage change of 24.2%, *p* < 0.007), pain assessment (percentage change of 66.6%, *p* < 0.001), first-line opioid of choice in children (percent change of 233.3%, *p* < 0.001), and palliative sedation (percent change of 51.9%, *p* < 0.001) had positive percent changes from the pre-test analysis that were statistically significant (Figure 1).

## 4. Discussion

This study, conducted in 22 hospital centers in several LMICs, described the development of an educational toolkit on PPC. Findings from this study reaffirmed the lack of knowledge in PC and pain management in children among HCPs and suggested that the workshops and the educational toolkit addressed both attitude and knowledge gaps of the participants. These educational strategies are beneficial tools for Latin American countries, with the potential to reduce the knowledge gap and promote the early integration of PC for high-risk pediatric populations.

Integration of PPC in the care of patients with life-threatening and life-limiting conditions: We found that most HCPs recognized that the integration of PPC care should ideally be done at the time of diagnosis of a life-threatening disease. This stands in contrast to previous reports that discussed significant barriers to achieve this objective [18]. If we all know that PC should be introduced early in the progression of the disease, why is it not being integrated and implemented? This is an important question to consider addressing prospectively and objectively through a multi-site study in Latin America.

Effective Communication: Communication is an essential tool in the care of patients with high medical complexity. Communication is critical in pediatrics because it must be adapted to each child’s cognitive and psychosocial development and the abilities of their parents or caregivers. We found that both physicians and other HCPs recognized the importance of effective and empathic communication. There is a gap regarding this topic in medical curricula [19] that must be addressed in the early stages of medical training as well as throughout other aspects of medical and clinical training.

Pain assessment in children: Organizations such as the Joint Commission International and the WHO have made pain assessment and management a priority in healthcare [20,21]. Initiatives include recording scores on pain assessment scales, quality improvement processes, and training healthcare personnel [22,23,24,25]. One of the most significant findings in our study was the lack of knowledge in addressing pain in children, and the greatest impact was in the acquisition of knowledge related to pain assessment in both physician and non-physician groups. The importance of this lies in the fact that pain assessment must be accurate and effective in order to manage pain. In other words, if we do not know how to assess pain, we cannot know how to treat it. This finding supports the idea of promoting the creation of hospital policies and educational spaces to reinforce knowledge in pain assessment for the pediatric population.

Opioids for the management of severe pain: The analysis results of this question were similar to that of the integration of PPC. If physicians dedicated to the care of children recognize that morphine is the opioid of choice in children, then why does the Colombian population have such a low per capita consumption of opioids [26,27]. We found that physicians and other HCPs recognized that morphine is the opioid of choice in the management of moderate to severe pain in children with life-threatening conditions [21]. However, misconceptions about using opioids persist even though it has been shown that by following a proper protocol, their use is considered to be effective and safe [28]. Development of institutional pain policies and guidelines and their promotion through educational initiatives is critical.

Palliative sedation at the end of life: A recent study on this subject concluded that there is ambiguity in the indication for palliative sedation. The pediatricians surveyed had varying interpretations of the concept and its ethical rationale [29]. This issue was clarified in our workshop and had a statistically significant percentage change after the workshop was implemented. The last workshop question asked the participants about the difference between palliative sedation and euthanasia; according to our results, there was clearly confusion about the concepts in the HCPs evaluated. We should seek out and implement educational strategies to clarify misconceptions on this topic. This is vitally important in Colombia as the only country in Latin America (one of the five countries globally) that has the authorization to perform euthanasia in children [30]. Likewise, it is crucial to deepen the knowledge about end-of-life care and better incorporate and integrate palliative care in children to address suffering and clarify these concepts as we work to promote dignified death in the pediatric population.

In the reported literature, we found several studies involving PPC education. They evaluated educational programs that encouraged compliance with the national clinical practice guideline for pediatric palliative care [31]; programs that supported and educated bedside staff on PPC to function as a liaison between them and the team [32]; programs that assessed the impact of the enhanced implementation of a palliative and end-of-life care educational curriculum for pediatrics [33]; and programs that assessed the development and dissemination of a palliative and end-of-life care educational curriculum in pediatrics [33].

Given the importance of the integration of PC in children with chronic and life-threatening diseases, it is essential to provide tools to approach the issues addressed in PPC, an objective achieved through the EmPalPed toolkit. The reviewed literature showed that no similar strategy exists in Latin America. Additionally, although the SARS-CoV-2 pandemic challenged us to implement strategies to continue this workshop virtually, it allowed us a much more extensive reach to remote areas of Colombia and other Latin American countries such as Peru and the Dominican Republic while still allowing for an effective intervention (there was no difference in findings before and after COVID). The implementation of this educational toolkit has required the initiation of specific strategies to promote awareness of and improve the education on PPC, which has been a difficult task. We hope this work can serve as a model for other regions and countries to contribute to the dissemination of PPC educational tools and materials.

Many of our findings were positive but should be evaluated with caution because we used an instrument that was not previously validated, although it was constructed by academic peers who are experts in the field, a nurse could not be present during all the interactive process of consensus-building of the workshop, and therefore may be a limitation. The virtual component of the toolkit was key to address the pandemic health restrictions but, could imply some delays or barriers during the development of the workshop such as internet connection, lack of focus and concentration, and lack of social interaction, with no comparison between the face-to-face method. A follow-up study is needed to measure the impact of the workshop on the quality of care of patients with palliative needs. Additionally, not all participants completed the pre-test and post-test. Finally, the evaluation was only a short-term follow-up, so further studies should be conducted to evaluate the impact of the workshop over more extended periods. It is also important to note that a change in knowledge does not necessarily mean a change in clinical practice, and objective outcome studies should follow.

## 5. Conclusions

The EmPalPed educational toolkit is a strategy that identifies and fills gaps in attitudes and knowledge about pediatric palliative care and pain management in health professionals in Latin America. Our evaluation demonstrated statistically significant short-term learning improvements on a variety of essential topics in pediatric palliative care. These tools should continue to be implemented to further reach the healthcare population providing care to suffering children and their families.

## Figures and Tables

**Figure 1 children-09-00838-f001:**
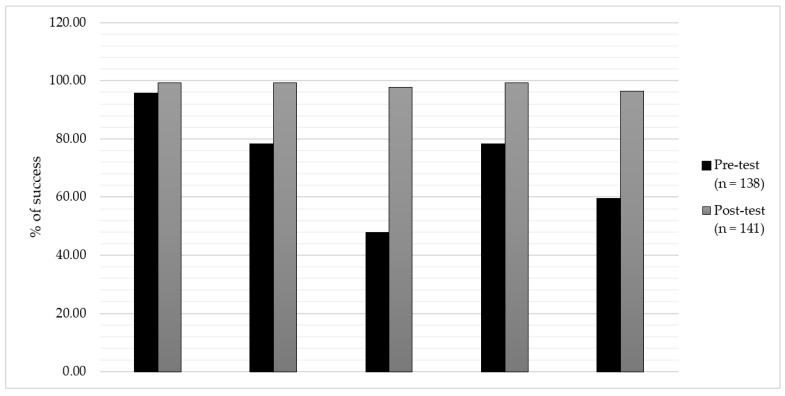
Evaluation of changes in areas before and after the intervention. PC, Palliative care.

**Table 1 children-09-00838-t001:** Characteristics of participants.

	PHYSICIANS *n* = 102		OTHER HCPs *n* = 43	
	*n*	%	*n*	%
Men	29	28.4	6	14.0
Women	73	71.6	37	86.0
**CLINICAL SPECIALTY**				
Pediatrician	50	34.5%		
General Practitioner	16	11.0%		
Pediatric Resident	8	5.5%		
Pediatric Intensive Care	4	2.8%		
Pediatric Oncologist	7	4.9%		
Neonatologist	4	2.8%		
Pediatric Neurologist	3	2.1%		
Pediatric Anesthesiologist	2	1.4%		
Pediatric Surgery	2	1.4%		
Family Medicine	2	1.4%		
Adult Pain and Palliative Care	1	0.7%		
Pediatric Cardiologist	1	0.7%		
Pediatric Gastroenterologist	1	0.7%		
Pediatric Pulmonologist	1	0.7%		
Nurse			15	10.3%
Psychologist			8	5.5%
Social Worker			8	5.5%
Other			12	8.3%

Others: occupational therapist, physical therapist, respiratory therapist, nurse assistants, speech therapists.

**Table 2 children-09-00838-t002:** Description of answers assessed between physicians and non-physicians in pre-test and post-test.

Areas	Pre-Test		Post-Test	
	Physicians (%) *n* = 96	Other Professional (%) *n* = 42	Physicians (%) *n* = 99	Other Professional (%) *n* = 42
Integration of PPC	94.8	97.6	100	97.6
Communication tools	78.1	78.6	100	97.6
Pain assessment	56.3	28.6	99	95.2
Opioid of choice for children	87.5	57.1	99	100
Palliative sedation	57.3	64.3	96	97.6

**Table 3 children-09-00838-t003:** Pre-test and post-test score of palliative care knowledge by area.

Areas	Physician						Other Professional					
	Pre-Test		Post-Test		% Change	*p* Value	Pre-Test		Post-Test		% Change	*p* Value
	*n* = 96		*n* = 99				*n* = 42		*n* = 42			
	*n* of Correct Answer	%	*n* of Correct Answer	%			*n* of Correct Answer	%	*n* of Correct Answer	%		
Integration PPC	91	94.8	99	100	8.8	*0.062*	41	97.6	41	97.6	0	*1*
Communication tools	75	78.1	99	100	32	<*0.001*	33	78.6	41	97.6	24.2	*0.007*
Pain assessment	54	56.3	98	99	81.5	<*0.001*	12	28.6	40	95.2	233.3	<0.001
Opioid of choice for children	84	87.5	98	99	16.7	<*0.001*	24	57.1	42	100	75	<0.001
Palliative sedation	55	57.3	95	96	72.7	<*0.001*	27	64.3	41	97.6	51.9	<0.001

## Data Availability

Data is contained within the article.

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
