# Peer review of "Using EmPalPed—An Educational Toolkit on Essential Messages in Palliative Care and Pain Management in Children—As a Strategy to Promote Pediatric Palliative Care"

_children, 2022, doi:10.3390/children9060838_

Round 1
Reviewer 1 Report
I applaud the initiative of the authors to apply and to study a strategy that improves the access to PC for children and their families with a specific tool.
The title announces a strategy to promote pediatric palliative care and the results present a simple but significant change of knowledge after giving an online workshop with a specific toolkit (i.e. EmPalPed). 145 HCPs from 22 hospitals in 12 workshops received online trainings and were asked to fill out a short, non-standardized questionnaire which was used as a pre-/post test.
Beside some typographical and spelling mistakes and formatting issues the text is well written and concise.
Major issue:
The link however, between the strategy and the results are methodologically weak.
The authors do neither present the content of the specific toolkit EmPalPed nor is it clear to the reader what the “strategy” is (to do teachings?) and how these would promote pediatric palliative care in general (as suggested by the title) or (as shown) specifically in Latin America. The applied method of a pre-/post-test of short-term memory knowledge does neither support the strategy nor validate the toolkit.
One possibility to improve the manuscript would be to propose either the mentioned strategy based on literature review and/or by introducing the applied toolkit EmPalPed maybe including the pre-/post-test as an add-on.
Minor issues:
Line 74: cards.Through
L 78: (Qo)
L 82: cinic
L 110/111: We found statistically significant differences were found
Table 2,3: formatting issues
Figure 1: Pre-test… Post-test…
L 180 confusión
Author Response
We thank the reviewer for pointing this out. We made suitable modifications to clarify the study methodology and added more details about the strategy and tools we used.
Reviewer 2 Report
Interesting study and work
Author Response
Thank you very much for your comment. We made a suitable revision to the English language and style.
Reviewer 3 Report
Thank you for a clearly presented and interesting article on a strategy to promote pediatric palliative care in Latin America through the development of EmPalPed. Despite all that has developed in pediatric palliative care in the past you have highlighted the need for much further work. The assessment of the situation, development of the Toolkit, identification of five key domains,the method of virtual classes in small groups, the reach across regions and countries and the statistically significant impact shown by the pre and post-test questionnaires were all clearly explained in your Tables. You identified the weaknesses and raised challenging questions most of which have global relevance and require answers. Your Bibliography was relevant and comprehensive and while there is a wealth of articles to select from your choice of references were relevant to the subject. Thank you for the opportunity to review your article and I look forward to reading of further development of pediatric palliative care in Colombia and throughout the region.
Author Response
We appreciate your observation and constructive comment. Thank you for supporting the development of pediatric palliative care in our region. We carry out a complete revision to the English language and style.